# Shear Strength of Trapezoidal Corrugated Steel Webs for Horizontally Curved Girder Bridges

**Sumei Liu [1,2]** , **Hanshan Ding [1,\*]** , **Luc Taerwe [2,3] and Wouter De Corte [2]**

[1]  School of Civil Engineering, Southeast University, Nanjing 210096, China; lsmf@seu.edu.cn
[2]  Department of Structural Engineering, Faculty of Engineering and Architecture, Ghent University,
    9000 Ghent, Belgium; Luc.Taerwe@UGent.be (L.T.); Wouter.DeCorte@UGent.be (W.D.C.)
[3]  College of Civil Engineering, Tongji University, Shanghai 200092, China
\*  Correspondence: hsding@seu.edu.cn

**Abstract:** Curved composite girder bridges with corrugated steel webs (CSWs) have already been constructed around the world. However, limited work has been done on their shear behavior. In this paper, the corrugated steel web (CSW) in horizontally curved girders (HCGs) is treated as an orthotropic cylindrical shallow shell, and the analytical formula for the elastic global shear buckling stress is deduced by the Galerkin method. Calculation tables for the global shear buckling coefficient for a four-edge simple support, for a four-edge fixed support, and for the two edges constrained by flanges fixed and the other two edges simply supported are given. Then, a parametric study based on a linear buckling analysis is performed to analyze the effect of the curvature radius and girder span on the shear buckling stress. Analytical and numerical results show that the difference of shear buckling stress of CSWs between curved girders and straight girders is small, so the shear design formulas for straight girders can be applied for curved girders. Finally, a series of tests were performed on three curved box girders with CSWs. Similar to CSWs in straight girders, the shear strain distributions of CSWs in HCGs are almost uniform along the direction of the web height and the principal strain direction angles are close to 45°. For the three specimens, CSWs carry about 76% of the shear force. In the destructive test, shear buckling after yielding occurred in all specimens which is in good agreement with the theoretical prediction, which means that the analytical formulas provide good predictions for the shear buckling stress of CSWs in HCGs and can be recommended for design purposes.

**Keywords:** horizontally curved girder; corrugated steel webs; shear buckling stress; Galerkin method; finite element analysis; experimental work

## 1. Introduction

The steel-concrete composite girder with CSWs is known as a new type of bridge structure to overcome the weight problem of common concrete box girders. Compared with concrete webs, CSWs have low longitudinal stiffness due to the accordion effect, so CSWs mainly carry the shear force and barely carry axial force [1]. Because of this characteristic, CSWs fail due to shear buckling or yielding. Therefore, the shear buckling stability of CSWs is one of the most important considerations in the design of this kind of composite girder bridges.

Curved composite girder bridges with CSWs (see Figure 1) have already been constructed around the world, for example, the Meaux viaduct in France, the Altwipfergrund viaduct in Germany, the Nakano viaduct in Japan, the Yuwotou bridge and No. 3 East River bridge in China, etc. However, so far, limited research has been conducted on the shear behavior of CSWs of HCGs. In practice, the shear buckling calculation of CSWs for HCGs usually adopts the corresponding formula of straight

girders which have been relatively well studied. It is widely accepted that for straight girder bridges, local shear buckling is the dominant failure mode in coarse corrugations, whereas global shear buckling becomes the dominant failure mode in dense corrugations and interactive shear buckling mode becomes dominant when the density is in between of the two above scenarios [2]. It is straightforward to understand that the shear buckling failure modes of CSWs in HCGs also consist of local shear buckling, global shear buckling, and interactive shear buckling.

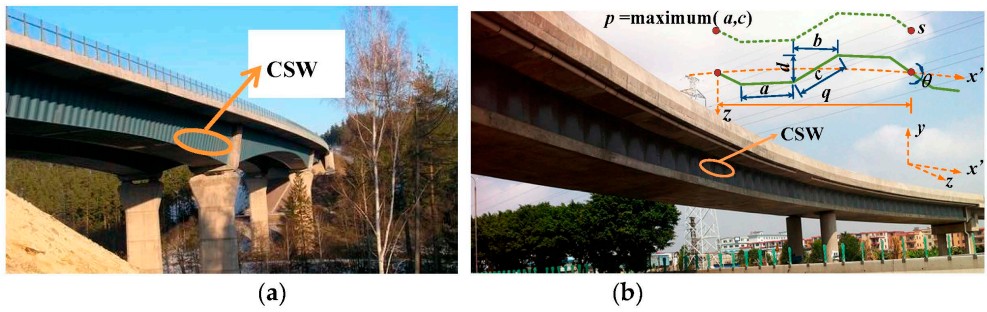

**Figure 1.** Curved composite girder bridges with corrugated steel webs (CSWs): (**a**) Altwipfergrund Viaduct; (**b**) Yuwotou Bridge. CSW: corrugated steel web.

The local shear buckling of CSWs for straight girders is solved by analyzing a single flat panel constrained by adjacent panels and girder flanges under shear force. For this, the formula for the shear buckling stress of isotropic rectangular plates [3] can be applied. Because the single flat panel of CSWs in HCGs is under the same constraint conditions as in straight girders, the formula can also be applied to HCGs. Aggarwal et al. [4] numerically studied the local shear buckling of CSWs and found that the boundary conditions at the top and bottom edges were close to fixed, while the boundary conditions at the fold line between the flat and inclined panels lie between simply supported and fixed.

The global shear buckling of CSWs for straight girder bridges is analyzed by assuming the whole web as an orthotropic rectangular plate constrained by concrete flanges and diaphragms, and has been studied by various researchers. Bergman and Reissner [5] derived the formula for calculating shear buckling loads by treating the corrugated plates as plates having different flexural rigidities in two perpendicular directions. Hlavacek [6] investigated the shear buckling behavior of stiffened plates reinforced by separate equally spaced stiffeners which were symmetrically arranged on two sides of the plates, and extended the deduced results to corrugated plates. Easley and McFarland [7] investigated the global shear buckling behavior of corrugated metal diaphragms also by treating them as orthotropic plates and developed the formulas for the shear buckling load by the Ritz and the Energy method. Easley [8] made a comparative analysis of the Bergmann-Reissner formula [5], the Hlavacek formula [6] and the Easley-McFarland formula [7], and proposed a more comprehensive and applicable global shear buckling formula of corrugated plates. Corrugated plates were originally applied in aircrafts and were gradually extended to civil engineering. The formula $\tau_g^e = k_g \frac{(D_x)^{1/4}(D_y)^{3/4}}{th^2}$ was accepted to calculate the global shear buckling stress of CSWs for straight girders, where $k_g$ is the global shear buckling coefficient depending on the edge conditions. Although a global shear buckling formula of CSWs has been proposed, researchers hold different views on the global shear buckling coefficient $k_g$. Easley [8] proposed $36 \leq k_g \leq 68.4$, 36 for a four-edge simple support and 68.4 for a four-edge fixed support, while Peterson [9] and Bergfelt et al. [10] suggested 32.4 for a four-edge simple support and 60.4 for a four-edge fixed support, and El Metwally and Loov [11] suggested 50 for composite girders with CSWs. The Guide to Stability Design Criteria for Metal Structures adopted 31.6 for a four-edge simple support and 59.2 for a four-edge fixed support [12]. Machimdamrong et al. [13] presented the transition curves of the elastic global shear buckling capacity from the case of a four-edge simple support to the case of a four-edge fixed support using the Rayleigh-Ritz method, but provides only the curves for the plate dimensions ($l \times h$) of 1 m $\times$ 1 m and 2 m $\times$ 1 m.

Finally, the interactive shear buckling formula for CSWs is determined by local shear buckling, global shear buckling and the yield stress of the plate material [14], but the way these parameters are to be combined is still the subject of debate. Important work has been done by Bergfelt et al. [10], El Metwally [15], Abbas et al. [16], Shiratani et al. [17], Sayed-Ahmed [18] and Yi et al. [14], etc., and various interactive shear buckling formulas of CSWs were proposed.

For practical applications, Elgaaly et al. [19] recommended that the capacity of CSWs was controlled by the smaller value of local and global buckling, and a semiempirical formula for the inelastic buckling stress when the elastic buckling stress is larger than 80% of the yield stress was proposed. Driver et al. [20] proposed a lower bound equation by combining local and global shear buckling equations of CSWs. Moon et al. [21] proposed a shear buckling parameter formula for trapezoidal CSWs with no need to calculate either local, global or interactive shear buckling parameters. Eldib [2] proposed a shear buckling parameter formula for curved CSWs. Nie et al. [22] carried out eight H-shape steel girders with CSWs and proposed a formula for the shear strength prediction of trapezoidal CSWs. Hassanein et al. investigated the shear behavior of linearly tapered bridge girders with CSWs [23], and high-strength steel corrugated web girders [24]. Leblouba and Barakat [25] experimentally and numerically studied the shear stress distribution in trapezoidal CSWs.

Basher et al. [26] studied the ultimate shear behavior of HCGs with trapezoidal CSWs and proposed an approximate method to calculate the shear strength of these girders. The ultimate shear strength of curved girders uses the equation of straight girders introducing a modification factor to account for the curvature of curved girders. Wang et al. [27] theoretically and numerically studied the elastic global shear buckling of HCGs with CSWs and proposed a global shear buckling stress formula for a four-edge simple support. From the studies mentioned above, it is clear that a lot of work has been done related to the shear behavior of straight girders with CSWs, but limited work to HCGs. The curvature of HCGs may indeed have an influence on the buckling of CSWs. Therefore, a comprehensive study is necessary to understand whether the shear buckling formulas for CSWs for straight girders are appropriate for HCGs, and if so, to which extent.

In this study, the CSW in HCGs is treated as an orthotropic cylindrical shallow shell constrained by the concrete flanges and diaphragms. First, the analytical formula for the elastic global shear buckling stress for three boundary conditions is deduced by the Galerkin method. Then, a parametric study based on a linear buckling analysis is performed to analyze the effect of the curvature radius and girder span on the shear buckling stress. Finally, a series of tests performed on three box girders with CSWs is discussed.

## 2. Elastic Global and Local Shear Buckling Stress of CSWs

### 2.1. Physical Equivalent Parameters of CSWs

For trapezoidal CSWs that are commonly used in actual girder bridges, when treated as an orthotropic plate or shell, the equivalent poisson's ratios $v_x$, $v_y$ [28], the equivalent elastic moduli $E_x$, $E_y$, the equivalent shear modulus $G_{xy}$ [29], the equivalent flexural stiffnesses $D_x$, $D_y$ and the torsional stiffness $D_{xy}$ per unit length of a CSW [7] can be expressed as Equations (1)–(8).

$$v_x = v \frac{E_x}{E} \tag{1}$$

$$v_y = v \tag{2}$$

$$E_x = \frac{t^2(a+b)}{d^2(3a+c)} E \tag{3}$$

$$E_y = \frac{s}{q} E \tag{4}$$

$$G_{xy} = \frac{q}{s} G \tag{5}$$

$$D_x = \frac{q}{s} \frac{Et^3}{12} \tag{6}$$

$$D_y = \frac{E(3a+c)td^2}{6q} \tag{7}$$

$$D_{xy} = \frac{s}{q}\frac{Et^3}{6(1+v)} \tag{8}$$

where $v$, $E$, $G$ are the poisson's ratio, the elastic modulus and the shear modulus of original steel plate respectively, and $t$ is the web thickness. As shown in Figure 1b, $a$ is the flat panel width; $b$ is the horizontal projection width of the inclined panel; $c$ is the inclined panel width; $d$ is the corrugation depth; $\theta$ is the corrugation angle; $q$ is the horizontal projection length of one periodic corrugation; $s$ is the total folded panel length of one periodic corrugation.

## 2.2. Elastic Global Shear Buckling Stress of CSWs

### 2.2.1. Critical Buckling Stress under Pure Shear

According to the theory of thin plates and shells, if the ratio of a shell's height to its short side is less than 0.2, the shell can be analyzed as a shallow shell. For HCGs with vertical CSWs, webs always meet this condition and can be treated as orthotropic cylindrical shallow shells (Figure 2) for the global shear buckling analysis.

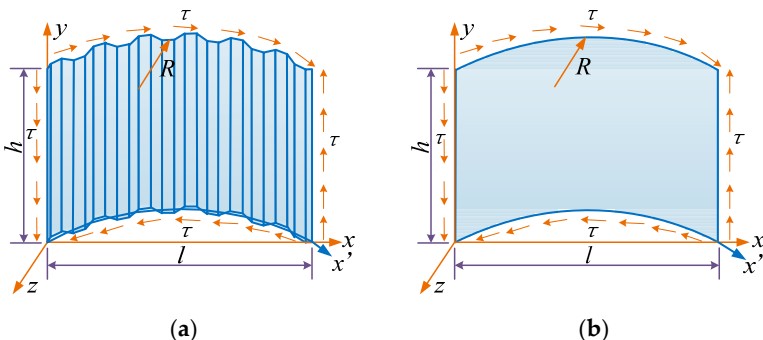

**Figure 2.** Corrugated steel web and its equivalent orthotropic cylindrical shallow shell: (**a**) Corrugated steel web; (**b**) Equivalent orthotropic cylindrical shallow shell.

According to the stability theory of plates and shells, the equilibrium equation and the deformation compatibility equation of an orthotropic cylindrical shallow shell shown in Figure 2 under pure shear force can be expressed respectively as Equations (9) and (10) [30]. In the following equations, the part that appears in bold type shows the difference between a plate and a shell.

$$\frac{1}{t}\left(D_x\frac{\partial^4}{\partial x^4} + D_{xy}\frac{\partial^4}{\partial x^2\partial y^2} + D_y\frac{\partial^4}{\partial y^4}\right)w + \mathbf{\frac{1}{R}\frac{\partial^2\Phi}{\partial y^2}} = 2\tau\frac{\partial^2 w}{\partial x\partial y} \tag{9}$$

$$\left[\frac{1}{E_y}\frac{\partial^4}{\partial x^4} + \left(\frac{1}{G_{xy}} - 2\frac{v_y}{E_y}\right)\frac{\partial^4}{\partial x^2\partial y^2} + \frac{1}{E_x}\frac{\partial^4}{\partial y^4}\right]\Phi - \mathbf{\frac{1}{R}\frac{\partial^2 w}{\partial y^2}} = 0 \tag{10}$$

where $\Phi$ is the stress function, $w$ is the out of plane deflection of the shell, $\tau$ is the shear stress, $R$ is the curvature radius of the HCG.

Substituting Equation (10) into Equation (9), Equation (9) can be expressed as Equation (11).

$$\frac{1}{t}\left(D_x\frac{\partial^4}{\partial x^4} + D_{xy}\frac{\partial^4}{\partial x^2\partial y^2} + D_y\frac{\partial^4}{\partial y^4}\right)w + \mathbf{\frac{1}{R^2}}\left[\frac{1}{E_y}\frac{\partial^4}{\partial x^4} + \left(\frac{1}{G_{xy}} - 2\frac{v_y}{E_y}\right)\frac{\partial^4}{\partial x^2\partial y^2} + \frac{1}{E_x}\frac{\partial^4}{\partial y^4}\right]^{-1}\mathbf{\frac{\partial^4 w}{\partial y^4}} = 2\tau\frac{\partial^2 w}{\partial x\partial y} \tag{11}$$

It can be assumed that the boundary conditions of CSWs satisfy a four-edge simple support, a four-edge fixed support, or the two edges constrained by flanges fixed and the other two edges simply supported (the edges $x = 0$ and $x = l$ are simply supported, the edges $y = 0$ and $y = h$ are fixed supported). The functions of deflection can be expressed respectively as Equations (12)–(14).

For a four-edge simple support [3]:

$$w = \sum_{m=1}^{\infty} \sum_{n=1}^{\infty} A_{mn} \sin \frac{m\pi x}{l} \sin \frac{n\pi y}{h} \tag{12}$$

For a four-edge fixed support [31]:

$$w = \sum_{m=1}^{\infty} \sum_{n=1}^{\infty} A_{mn} \left[ \frac{1}{m} \sin \frac{m\pi x}{l} - \frac{1}{m+2} \sin \frac{(m+2)\pi x}{l} \right] \left[ \frac{1}{n} \sin \frac{n\pi y}{h} - \frac{1}{n+2} \sin \frac{(n+2)\pi y}{h} \right] \tag{13}$$

For the edges $x = 0$ and $x = l$ simply supported, and the edges $y = 0$ and $y = h$ fixed:

$$w = \sum_{m=1}^{\infty} \sum_{n=1}^{\infty} A_{mn} \sin \frac{m\pi x}{l} \left[ \frac{1}{n} \sin \frac{n\pi y}{h} - \frac{1}{n+2} \sin \frac{(n+2)\pi y}{h} \right] \tag{14}$$

where $h$ is the web height equal to the clear distance between the top and bottom concrete flanges, $l$ is the web length equal to the linear distance between the two adjacent diaphragm plates.

By substituting Equations (12)–(14) into Equation (11), defining $\alpha = \frac{D_x}{D_y} = \frac{E_x}{E_y}$, $\beta = \frac{D_{xy}}{D_y}$, $\gamma = \frac{G_{xy}}{E_y - 2\nu_y G_{xy}}$, $\lambda = l/h$, considering $\frac{E_y}{D_y} = \frac{6s}{td^2(3a+c)}$, and according to the Galerkin method, Equation (11) can be simplified as Equations (15)–(17) respectively.

For the four-edge simple support:

$$\frac{D_y}{4th^2} \left[ \frac{\pi^4}{\lambda^3} \left( \alpha m^4 + \beta \lambda^2 m^2 n^2 + \lambda^4 n^4 \right) + \alpha \gamma \lambda^5 n^4 \left( \alpha \gamma m^4 + \alpha \lambda^2 m^2 n^2 + \gamma \lambda^4 n^4 \right)^{-1} \frac{h^4}{R^2 d^2} \frac{6s}{(3a+c)} \right] A_{mn}$$
$$-8\tau \sum_{i}^{\infty} \sum_{j}^{\infty} A_{ij} \frac{mnij}{(m^2-i^2)(n^2-j^2)} = 0 \, (m \pm i = \text{odd}, n \pm j = \text{odd}) \tag{15}$$

For the four-edge fixed support:

$$
\begin{aligned}
&\frac{D_y}{th^2} \frac{\pi^4}{4\lambda^3} \left\{
\begin{array}{l}
A_{mn}\left\{\alpha\left[m^2 + (m+2)^2\right]\left[n^{-2} + (n+2)^{-2}\right] + 4\beta\lambda^2 + \lambda^4\left[m^{-2} + (m+2)^{-2}\right]\left[n^2 + (n+2)^2\right]\right\} \\
-A_{m,n+2}\left\{\alpha\left[m^2 + (m+2)^2\right](n+2)^{-2} + 2\beta\lambda^2 + \lambda^4\left[m^{-2} + (m+2)^{-2}\right](n+2)^2\right\} \\
-A_{m,n-2}\left\{\alpha\left[m^2 + (m+2)^2\right]n^{-2} + 2\beta\lambda^2 + \lambda^4\left[m^{-2} + (m+2)^{-2}\right]n^2\right\} \\
-A_{m+2,n}\left\{\alpha(m+2)^2\left[n^{-2} + (n+2)^{-2}\right] + 2\beta\lambda^2 + \lambda^4(m+2)^{-2}\left[n^2 + (n+2)^2\right]\right\} \\
+A_{m+2,n+2}\left[\alpha(m+2)^2(n+2)^{-2} + \beta\lambda^2 + \lambda^4(m+2)^{-2}(n+2)^2\right] \\
+A_{m+2,n-2}\left[\alpha(m+2)^2n^{-2} + \beta\lambda^2 + \lambda^4(m+2)^{-2}n^2\right] \\
-A_{m-2,n}\left\{\alpha m^2\left[n^{-2} + (n+2)^{-2}\right] + 2\beta\lambda^2 + \lambda^4 m^{-2}\left[n^2 + (n+2)^2\right]\right\} \\
+A_{m-2,n+2}\left[\alpha m^2(n+2)^{-2} + \beta\lambda^2 + \lambda^4 m^{-2}(n+2)^2\right] \\
+A_{m-2,n-2}\left[\alpha m^2 n^{-2} + \beta\lambda^2 + \lambda^4 m^{-2}n^2\right]
\end{array}
\right\} \\[4pt]
&+\frac{D_y}{th^2} \frac{h^4}{R^2 d^2} \frac{6s}{(3a+c)} \frac{\alpha\gamma\lambda^5}{4} \left\{
\begin{array}{l}
A_{mn}\left\{
\begin{array}{l}
\left(\alpha\gamma m^4 + \alpha\lambda^2 m^2 n^2 + \gamma\lambda^4 n^4\right)^{-1}m^{-2}n^2 \\
+\left[\alpha\gamma m^4 + \alpha\lambda^2 m^2(n+2)^2 + \gamma\lambda^4(n+2)^4\right]^{-1}m^{-2}(n+2)^2 \\
+\left[\alpha\gamma(m+2)^4 + \alpha\lambda^2(m+2)^2 n^2 + \gamma\lambda^4 n^4\right]^{-1}(m+2)^{-2}n^2 \\
+\left[\alpha\gamma(m+2)^4 + \alpha\lambda^2(m+2)^2(n+2)^2 + \gamma\lambda^4(n+2)^4\right]^{-1}(m+2)^{-2}(n+2)^2
\end{array}
\right\} \\
-A_{m,n+2}\left\{
\begin{array}{l}
\left[\alpha\gamma m^4 + \alpha\lambda^2 m^2(n+2)^2 + \gamma\lambda^4(n+2)^4\right]^{-1}m^{-2}(n+2)^2 \\
+\left[\alpha\gamma(m+2)^4 + \alpha\lambda^2(m+2)^2(n+2)^2 + \gamma\lambda^4(n+2)^4\right]^{-1}(m+2)^{-2}(n+2)^2
\end{array}
\right\} \\
-A_{m,n-2}\left\{
\begin{array}{l}
\left(\alpha\gamma m^4 + \alpha\lambda^2 m^2 n^2 + \gamma\lambda^4 n^4\right)^{-1}m^{-2}n^2 \\
+\left[\alpha\gamma(m+2)^4 + \alpha\lambda^2(m+2)^2 n^2 + \gamma\lambda^4 n^4\right]^{-1}(m+2)^{-2}n^2
\end{array}
\right\} \\
-A_{m+2,n}\left\{
\begin{array}{l}
\left[\alpha\gamma(m+2)^4 + \alpha\lambda^2(m+2)^2 n^2 + \gamma\lambda^4 n^4\right]^{-1}(m+2)^{-2}n^2 \\
+\left[\alpha\gamma(m+2)^4 + \alpha\lambda^2(m+2)^2(n+2)^2 + \gamma\lambda^4(n+2)^4\right]^{-1}(m+2)^{-2}(n+2)^2
\end{array}
\right\} \\
+A_{m+2,n+2}\left[\alpha\gamma(m+2)^4 + \alpha\lambda^2(m+2)^2(n+2)^2 + \gamma\lambda^4(n+2)^4\right]^{-1}(m+2)^{-2}(n+2)^2 \\
+A_{m+2,n-2}\left[\alpha\gamma(m+2)^4 + \alpha\lambda^2(m+2)^2 n^2 + \gamma\lambda^4 n^4\right]^{-1}(m+2)^{-2}n^2 \\
-A_{m-2,n}\left\{
\begin{array}{l}
\left(\alpha\gamma m^4 + \alpha\lambda^2 m^2 n^2 + \gamma\lambda^4 n^4\right)^{-1}m^{-2}n^2 \\
+\left[\alpha\gamma m^4 + \alpha\lambda^2 m^2(n+2)^2 + \gamma\lambda^4(n+2)^4\right]^{-1}m^{-2}(n+2)^2
\end{array}
\right\} \\
+A_{m-2,n+2}\left[\alpha\gamma m^4 + \alpha\lambda^2 m^2(n+2)^2 + \gamma\lambda^4(n+2)^4\right]^{-1}m^{-2}(n+2)^2 \\
+A_{m-2,n-2}\left(\alpha\gamma m^4 + \alpha\lambda^2 m^2 n^2 + \gamma\lambda^4 n^4\right)^{-1}m^{-2}n^2
\end{array}
\right\} \\[4pt]
&-8\tau \sum_{i=1}^{\infty} \sum_{j=1}^{\infty} A_{ij} \left\{
\begin{array}{l}
\left[\frac{1}{m^2-i^2} - \frac{1}{(m+2)^2-i^2} - \frac{1}{m^2-(i+2)^2} + \frac{1}{(m+2)^2-(i+2)^2}\right] \\
\times\left[\frac{1}{n^2-j^2} - \frac{1}{(n+2)^2-j^2} - \frac{1}{n^2-(j+2)^2} + \frac{1}{(n+2)^2-(j+2)^2}\right]
\end{array}
\right\} = 0
\end{aligned}
\tag{16}
$$

For the edges $x = 0$ and $x = l$ simply supported, and the edges $y = 0$ and $y = h$ fixed:

$$
\begin{aligned}
&\frac{D_y}{th^2}\frac{\pi^4}{4\lambda^3}\left\{
\begin{array}{l}
A_{mn}\left\{\alpha m^4\left[n^{-2} + (n+2)^{-2}\right] + 2\beta\lambda^2 m^2 + \lambda^4\left[n^2 + (n+2)^2\right]\right\} \\
-A_{m,n+2}\left[\alpha m^4(n+2)^{-2} + \beta\lambda^2 m^2 + \lambda^4(n+2)^2\right] - A_{m,n-2}\left[\alpha m^4 n^{-2} + \beta\lambda^2 m^2 + \lambda^4 n^2\right]
\end{array}
\right\} \\
&+\frac{D_y}{th^2}\frac{h^4}{R^2d^2}\frac{6s}{(3a+c)}\frac{\alpha\gamma\lambda^5}{4}
\left\{
\begin{array}{l}
A_{mn}\left\{
\begin{array}{l}
\left(\alpha\gamma m^4 + \alpha\lambda^2 m^2 n^2 + \gamma\lambda^4 n^4\right)^{-1}n^2 \\
+\left[\alpha\gamma m^4 + \alpha\lambda^2 m^2(n+2)^2 + \gamma\lambda^4(n+2)^4\right]^{-1}(n+2)^2
\end{array}
\right\} \\
-A_{m,n+2}\left[\alpha\gamma m^4 + \alpha\lambda^2 m^2(n+2)^2 + \gamma\lambda^4(n+2)^4\right]^{-1}(n+2)^2 \\
-A_{m,n-2}\left(\alpha\gamma m^4 + \alpha\lambda^2 m^2 n^2 + \gamma\lambda^4 n^4\right)^{-1}n^2
\end{array}
\right\} \\
&-8\tau\sum_{i=1}^{\infty}\sum_{j=1}^{\infty}A_{ij}\frac{mi}{m^2-i^2}\left[\frac{1}{n^2-j^2} - \frac{1}{(n+2)^2-j^2} - \frac{1}{n^2-(j+2)^2} + \frac{1}{(n+2)^2-(j+2)^2}\right] = 0
\end{aligned}
\tag{17}
$$

By assigning values to $m$ and $n$ in Equations (15)–(17), a series of linear algebraic equations with $A_{ij}$ as unknowns can be obtained. Then the critical shear buckling stress can be derived by assuming the coefficient determinant of the linear algebraic equations equals zero. (i.e., a linear bifurcation analysis).

According to Equations (15)–(17), the elastic global shear buckling stress of CSWs can be expressed as Equation (18):

$$
\tau_g^e = k_g\frac{D_y}{h^2 t}
\tag{18}
$$

where $k_g$ is the elastic global shear buckling coefficient of CSWs in HCGs. The detailed solution process of the coefficient $k_{g,s}$ for a four-edge simple support, $k_{g,f}$ for a four-edge fixed support, $k_{g,fs}$ for the edges $x = 0$ and $x = l$ simply supported, and the edges $y = 0$ and $y = h$ fixed is given below.

### 2.2.2. Calculation of Coefficient $k_g$

According to Equations (15)–(18), the global shear buckling coefficient of CSWs in HCGs $k_g$ is associated with the length to height ratio $\lambda$ ($l/h$), the rigidity ratios $\alpha(D_x/D_y)$ and $\beta(D_{xy}/D_y)$, the modulus ratio $\gamma(G_{xy}/(E_y - 2\nu_y G_{xy}))$, the expression $h^2/(Rd)$ and $6s/(3a + c)$.

1. Value ranges of $\alpha$, $\beta$, $\gamma$, $h^2/(Rd)$ and $6s/(3a + c)$ for common HCG bridges with CSWs

For trapezoidal CSWs that are commonly used in girder bridges, the rigidity ratios $\alpha$ and $\beta$ have the relationship: $\beta/\alpha = 2s^2/[(1 + \nu)q^2]$. A statistical analysis of available bridges with CSWs (as shown in Table 1) shows that the rigidity ratio $\alpha$ varies from 0.0006 to 0.0069, $\gamma$ varies from 0.38 to 0.44, $6s/(3a + c)$ is about 6, and $\beta$ is about $(1.67\sim2.0)\alpha$. The following parametric study considers $\alpha$ ranging from 0.0005 to 0.0070, $\gamma$ ranging from 0.38 to 0.44, $6s/(3a + c)$ equal to 6 and $\beta$ equal to $1.6\alpha$, $1.8\alpha$, $2.0\alpha$ respectively.

**Table 1.** The geometry of CSWs in available bridges.

| Bridges | $a$ | $c$ | $d$ | $t_{min}$ | $t_{max}$ | $\frac{6s}{3a+c}$ | Corresponding to $t_{min}$ | | | Corresponding to $t_{max}$ | | | $\gamma$ |
|---|---|---|---|---|---|---|---|---|---|---|---|---|---|
| | mm | mm | mm | mm | mm | | $\alpha$ | $\beta$ | $\beta/\alpha$ | $\alpha$ | $\beta$ | $\beta/\alpha$ | |
| Cognac | 353 | 353 | 150 | 8 | 8 | 6.00 | 0.0013 | 0.0022 | 1.69 | 0.0013 | 0.0022 | 1.69 | 0.4406 |
| Maupre | 284 | 284 | 150 | 8 | 8 | 6.00 | 0.0012 | 0.0022 | 1.83 | 0.0012 | 0.0022 | 1.83 | 0.4093 |
| Dole | 430 | 430 | 220 | 8 | 12 | 6.00 | 0.0006 | 0.0010 | 1.67 | 0.0013 | 0.0023 | 1.77 | 0.4159 |
| Shinkai | 250 | 250 | 150 | 9 | 9 | 6.00 | 0.0015 | 0.0028 | 1.87 | 0.0015 | 0.0028 | 1.87 | 0.3832 |
| Miyukibashi | 300 | 300 | 150 | 8 | 12 | 6.00 | 0.0012 | 0.0022 | 1.83 | 0.0028 | 0.0049 | 1.75 | 0.4193 |
| Katsutegawa | 430 | 430 | 220 | 9 | 12 | 6.00 | 0.0007 | 0.0013 | 1.86 | 0.0013 | 0.0023 | 1.77 | 0.4159 |
| Hontani | 330 | 336 | 200 | 9 | 14 | 6.03 | 0.0008 | 0.0016 | 2.00 | 0.0020 | 0.0038 | 1.90 | 0.3841 |
| Koinumarukawa | 430 | 430 | 220 | 9 | 16 | 6.00 | 0.0007 | 0.0013 | 1.86 | 0.0023 | 0.0041 | 1.78 | 0.4159 |
| Shimoda | 430 | 430 | 220 | 12 | 16 | 6.00 | 0.0013 | 0.0023 | 1.77 | 0.0023 | 0.0041 | 1.78 | 0.4159 |
| Nakano Viaduct | 330 | 336 | 200 | 9 | 19 | 6.03 | 0.0008 | 0.0016 | 2.00 | 0.0037 | 0.0069 | 1.86 | 0.3841 |
| Kurobekawa Railway | 400 | 400 | 200 | 12 | 25 | 6.00 | 0.0016 | 0.0028 | 1.75 | 0.0069 | 0.0120 | 1.74 | 0.4240 |
| Altwipfergrund | 360 | 360 | 220 | 10 | 22 | 6.00 | 0.0008 | 0.0016 | 2.00 | 0.0041 | 0.0077 | 1.88 | 0.3832 |
| Juancheng-Huanghe | 430 | 421 | 200 | 10 | 18 | 5.97 | 0.0011 | 0.0019 | 1.73 | 0.0036 | 0.0062 | 1.72 | 0.4270 |
| Henan-Pohe | 250 | 250 | 150 | 8 | 8 | 6.00 | 0.0012 | 0.0022 | 1.83 | 0.0012 | 0.0022 | 1.83 | 0.3832 |
| Wei River | 330 | 336 | 200 | 8 | 12 | 6.03 | 0.0007 | 0.0012 | 1.71 | 0.0015 | 0.0028 | 1.87 | 0.3841 |
| Nanjing-Chuhe | 430 | 430 | 220 | 10 | 18 | 6.00 | 0.0009 | 0.0016 | 1.78 | 0.0029 | 0.0051 | 1.76 | 0.4159 |

Note: $t_{max}$ and $t_{min}$ are the maximum and minimum thicknesses of CSWs respectively when an available bridge has more than one thickness value.

Table 1 shows that the corrugation depth $d$ varies from 0.15 m to 0.22 m. The upper limit of $H/d$ does not exceed 133 considering the girder height $H$ generally is not more than 20 m. For common HCG bridges, the girder height to length ratio $H/L$ generally ranges from 1/11 to 1/30, and the central angle $L/R$ is generally no more than $\pi/2$. So, the upper limit of $H/R$ of HCG bridges does not exceed 0.143, where $L$ is the girder span. Thus, the upper limit of $H^2/(Rd)$ of HCG bridges does not exceed 20. Because the CSW height $h$ is smaller than the girder height $H$, the upper limit of $h^2/(Rd)$ does not exceed 20. In order to expand the scope of application of the formulas, the following parametric study considers $h^2/(Rd)$ ranging from 0 to 30.

2.　Influence of $\alpha$ and $\beta/\alpha$ on the global shear buckling coefficient $k_g$

Theoretically, the more numbers used in the trigonometric series (as shown in Equations (12)–(14)), the more precise the solution is. If $m$ and $n$ increase toward infinity, exact results of shear buckling stress of CSWs can be obtained. However, the calculation effort increases with the increasing numbers $m$ and $n$ in the trigonometric series. In the case of the CSW with a length to height ratio $l/h$ less than 5, the deviation between the results with $m = 30$, $n = 30$ and the results with $m = 25$, $n = 25$ is less than 1%. In what follows, $m = 30$ and $n = 30$ are adopted.

Table 2 shows the values of $k_g$ calculated for various values of $D_x/D_y$ and $l/h$, and for $\beta = 1.6\alpha$, $\beta = 1.8\alpha$ and $\beta = 2.0\alpha$ respectively when $h^2/(Rd) = 5$ and $\gamma = 0.4$ for a four-edge simple support. The results for $\beta = 1.6\alpha$ and $\beta = 2.0\alpha$, compared to for $\beta = 1.8\alpha$, deviate less than 0.6%. The results show that the parameter $\beta/\alpha$ has little effect on the coefficient $k_g$ for common bridges with CSWs. From an engineering application point of view, the deviations can be ignored. In addition, the conclusion remains unchanged when changing the values of $h^2/(Rd)$, $\gamma$, and the boundary conditions. As a result, $\beta = 1.8\alpha$ is used further in this paper.

**Table 2.** The effect of $\beta/\alpha$ on the global shear buckling coefficient $k_{g,s}$ when $h^2/(Rd) = 5$ and $\gamma = 0.4$ for the four-edge simple support.

| $D_x/D_y$ | | $l/h$ | | | | |
|---|---|---|---|---|---|---|
| | | 1 | 2 | 3 | 4 | 5 |
| 0.0005 | $\beta = 1.6\alpha$ | 5.0169 | 4.9471 | 4.9335 | 4.9292 | 4.9252 |
| | $\beta = 1.8\alpha$ | 5.0245 | 4.9547 | 4.9410 | 4.9367 | 4.9327 |
| | $\beta = 2.0\alpha$ | 5.0321 | 4.9623 | 4.9486 | 4.9442 | 4.9402 |
| 0.0015 | $\beta = 1.6\alpha$ | 6.7307 | 6.5947 | 6.5643 | 6.5525 | 6.5470 |
| | $\beta = 1.8\alpha$ | 6.7489 | 6.6121 | 6.5816 | 6.5697 | 6.5642 |
| | $\beta = 2.0\alpha$ | 6.7671 | 6.6295 | 6.5989 | 6.5868 | 6.5813 |
| 0.0025 | $\beta = 1.6\alpha$ | 7.7454 | 7.5645 | 7.5126 | 7.4963 | 7.4888 |
| | $\beta = 1.8\alpha$ | 7.7714 | 7.5901 | 7.5379 | 7.5214 | 7.5139 |
| | $\beta = 2.0\alpha$ | 7.7973 | 7.6151 | 7.5631 | 7.5465 | 7.5390 |
| 0.0035 | $\beta = 1.6\alpha$ | 8.5136 | 8.2755 | 8.2214 | 8.2008 | 8.1909 |
| | $\beta = 1.8\alpha$ | 8.5485 | 8.3084 | 8.2539 | 8.2332 | 8.2232 |
| | $\beta = 2.0\alpha$ | 8.5834 | 8.3413 | 8.2865 | 8.2656 | 8.2555 |
| 0.0050 | $\beta = 1.6\alpha$ | 9.5393 | 9.1230 | 9.0567 | 9.0301 | 9.0159 |
| | $\beta = 1.8\alpha$ | 9.5816 | 9.1656 | 9.0993 | 9.0721 | 9.0580 |
| | $\beta = 2.0\alpha$ | 9.6239 | 9.2081 | 9.1418 | 9.1141 | 9.1001 |
| 0.0070 | $\beta = 1.6\alpha$ | 10.4099 | 10.0444 | 9.9332 | 9.8997 | 9.8847 |
| | $\beta = 1.8\alpha$ | 10.4672 | 10.1004 | 9.9878 | 9.9537 | 9.9386 |
| | $\beta = 2.0\alpha$ | 10.5245 | 10.1557 | 10.0424 | 10.0077 | 9.9924 |

Figure 3 shows the effect of the rigidity ratio $D_x/D_y$ and the length to height ratio $l/h$ on the global shear buckling coefficient $k_{g,s}$ for a four-edge simple support. As we can see from Figure 3, the global shear buckling coefficient $k_{g,s}$ increases with the increase of the rigidity ratio $D_x/D_y$ and decreases with the increase of the length to height ratio $l/h$ but only very little. When $l/h$ is larger than

2, which is common for bridges, the change of $k_{g,s}$ is minimal and the values of $k_{g,s}$ show a converging trend. The conclusion remains unchanged when changing the values of $h^2/(Rd)$, $\gamma$, and the boundary conditions. Because the values of $k_g$ show a converging trend when $l/h$ is larger than 2, assuming $l/h = 5$ for further calculation will not only ensure the accuracy of the calculation but also meet the engineering requirements of design simplicity.

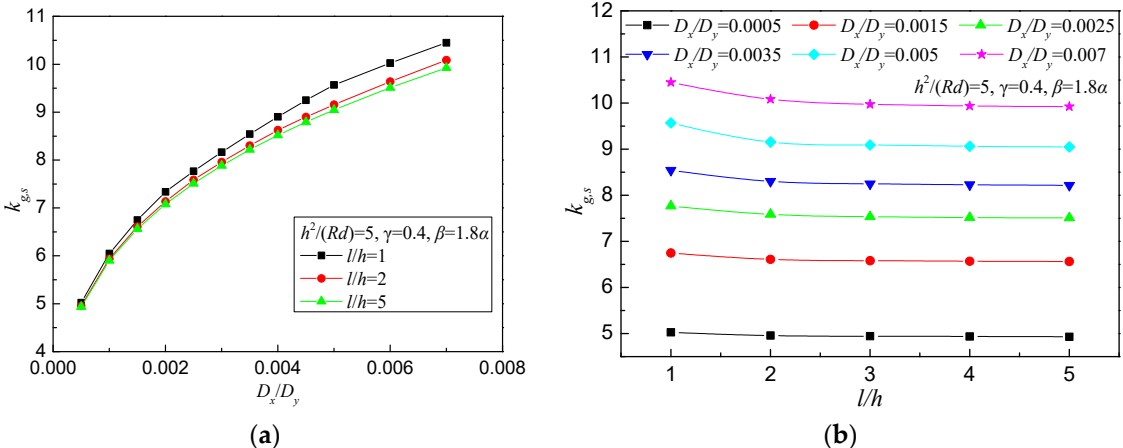

**Figure 3.** The effect of the rigidity ratio $D_x/D_y$ and the length to height ratio $l/h$ on the global shear buckling coefficient $k_{g,s}$ for the four-edge simple support: (**a**) $D_x/D_y$ on $k_{g,s}$; (**b**) $l/h$ on $k_{g,s}$.

3.　Influence of $\gamma$ on the global shear buckling coefficient $k_g$

Table 3 lists the values of $k_{g,s}$ for $\gamma$ equal to 0.38, 0.4, 0.42 and 0.44 respectively when $l/h = 5$, $h^2/(Rd)$ = 5 and $\beta = 1.8\alpha$ for the four-edge simple support. The results are practically equal for $\gamma$ equal to 0.38, 0.4, 0.42 and 0.44. The conclusion remains unchanged when changing the values of $l/h$, $h^2/(Rd)$, and the boundary conditions. This implies that the modulus ratio $\gamma$ has little effect on the coefficient $k_g$ for common HCG bridges with CSWs. In what follows, assuming $\gamma = 0.4$ will not only ensure the accuracy of the calculation but also reduce the number of parameters in the parametric study.

**Table 3.** The effect of $\gamma$ on the global shear buckling coefficient $k_{g,s}$ when $l/h = 5$, $h^2/(Rd) = 5$ and $\beta = 1.8\alpha$ for the four-edge simple support.

| $\alpha$ | $\gamma = 0.38$ | $\gamma = 0.40$ | $\gamma = 0.42$ | $\gamma = 0.44$ |
|---|---|---|---|---|
| $\alpha = 0.001$ | 5.9044 | 5.9044 | 5.9044 | 5.9044 |
| $\alpha = 0.003$ | 7.8894 | 7.8894 | 7.8894 | 7.8894 |
| $\alpha = 0.005$ | 9.0580 | 9.0580 | 9.0581 | 9.0581 |

4.　Influence of $h^2/(Rd)$ on the global shear buckling coefficient $k_g$

Tables 4–6 lists the values of $k_g$ for various values of $h^2/(Rd)$ and $D_x/D_y$ when $l/h = 5$, $\gamma = 0.4$ and $\beta = 1.8\alpha$ for three boundary conditions. Figure 4 shows the effect of $h^2/(Rd)$ and $D_x/D_y$ on the global shear buckling coefficient $k_{g,s}$ for the four-edge simple support. As we can see from Figure 4 and Tables 4–6, the global shear buckling coefficient $k_g$ increases with the increase of the rigidity ratio $D_x/D_y$ and increases slightly with the parameter $h^2/(Rd)$. This implies that the global shear buckling stress of curved bridge webs is slightly higher than that of straight bridge webs. For HCG bridges with $D_x/D_y \leq 0.007$ and $h^2/(Rd) \leq 20$, the global shear buckling stress of curved bridge webs can be calculated conservatively as that of straight bridge webs with a difference less than 2.5%.

**Table 4.** Global shear buckling coefficient $k_{g,s}$ for various values of $h^2/(Rd)$ and $D_x/D_y$ when $l/h = 5$, $\gamma = 0.4$ and $\beta = 1.8\alpha$ for the four-edge simple support.

| $h^2/(Rd)$ | $D_x/D_y$ | | | | | | | | | | | |
|---|---|---|---|---|---|---|---|---|---|---|---|---|
| | 0.0005 | 0.001 | 0.0015 | 0.002 | 0.0025 | 0.003 | 0.0035 | 0.004 | 0.0045 | 0.005 | 0.006 | 0.007 |
| 0 | 4.9321 | 5.9031 | 6.5619 | 7.0786 | 7.5097 | 7.8841 | 8.2169 | 8.5171 | 8.7936 | 9.0482 | 9.5103 | 9.9235 |
| 5 | 4.9327 | 5.9044 | 6.5642 | 7.0818 | 7.5139 | 7.8894 | 8.2232 | 8.5246 | 8.8024 | 9.0580 | 9.5226 | 9.9386 |
| 10 | 4.9344 | 5.9085 | 6.5708 | 7.0912 | 7.5266 | 7.9052 | 8.2419 | 8.5468 | 8.8283 | 9.0872 | 9.5592 | 9.9824 |
| 15 | 4.9373 | 5.9153 | 6.5819 | 7.1068 | 7.5473 | 7.9309 | 8.2726 | 8.5832 | 8.8698 | 9.1348 | 9.6188 | 10.0522 |
| 20 | 4.9414 | 5.9247 | 6.5973 | 7.1283 | 7.5757 | 7.9658 | 8.3147 | 8.6331 | 8.9261 | 9.1996 | 9.6983 | 10.1463 |
| 25 | 4.9465 | 5.9368 | 6.6167 | 7.1555 | 7.6110 | 8.0096 | 8.3675 | 8.6951 | 8.9962 | 9.2792 | 9.7946 | 10.2621 |
| 30 | 4.9527 | 5.9515 | 6.6399 | 7.1881 | 7.6532 | 8.0616 | 8.4300 | 8.7665 | 9.0788 | 9.3706 | 9.9070 | 10.3954 |

**Table 5.** Global shear buckling coefficient $k_{g,f}$ for various values of $h^2/(Rd)$ and $D_x/D_y$ when $l/h = 5$, $\gamma = 0.4$ and $\beta = 1.8\alpha$ for the four-edge fixed support.

| $h^2/(Rd)$ | $D_x/D_y$ | | | | | | | | | | | |
|---|---|---|---|---|---|---|---|---|---|---|---|---|
| | 0.0005 | 0.001 | 0.0015 | 0.002 | 0.0025 | 0.003 | 0.0035 | 0.004 | 0.0045 | 0.005 | 0.006 | 0.007 |
| 0 | 9.3520 | 11.1647 | 12.4023 | 13.3594 | 14.1630 | 14.8562 | 15.4733 | 16.0290 | 16.5382 | 17.0068 | 17.8556 | 18.6097 |
| 5 | 9.3521 | 11.1650 | 12.4028 | 13.3602 | 14.1640 | 14.8574 | 15.4748 | 16.0308 | 16.5402 | 17.0091 | 17.8585 | 18.6132 |
| 10 | 9.3525 | 11.1659 | 12.4044 | 13.3624 | 14.1669 | 14.8611 | 15.4793 | 16.0360 | 16.5463 | 17.0161 | 17.8672 | 18.6237 |
| 15 | 9.3532 | 11.1675 | 12.4071 | 13.3661 | 14.1718 | 14.8672 | 15.4867 | 16.0447 | 16.5564 | 17.0276 | 17.8817 | 18.6411 |
| 20 | 9.3541 | 11.1697 | 12.4109 | 13.3713 | 14.1786 | 14.8758 | 15.4971 | 16.0570 | 16.5706 | 17.0437 | 17.9019 | 18.6653 |
| 25 | 9.3554 | 11.1725 | 12.4157 | 13.3780 | 14.1874 | 14.8868 | 15.5103 | 16.0726 | 16.5887 | 17.0643 | 17.9276 | 18.6963 |
| 30 | 9.3568 | 11.1760 | 12.4216 | 13.3861 | 14.1990 | 14.8998 | 15.5264 | 16.0917 | 16.6108 | 17.0894 | 17.9587 | 18.7340 |

**Table 6.** Global shear buckling coefficient $k_{g,fs}$ for various values of $h^2/(Rd)$ and $D_x/D_y$ when $l/h = 5$, $\gamma = 0.4$ and $\beta = 1.8\alpha$ for the two edges constrained by flanges fixed and the other two edges simply supported.

| $h^2/(Rd)$ | $D_x/D_y$ | | | | | | | | | | | |
|---|---|---|---|---|---|---|---|---|---|---|---|---|
| | 0.0005 | 0.001 | 0.0015 | 0.002 | 0.0025 | 0.003 | 0.0035 | 0.004 | 0.0045 | 0.005 | 0.006 | 0.007 |
| 0 | 9.3514 | 11.1640 | 12.4009 | 13.3583 | 14.1615 | 14.8549 | 15.4721 | 16.0277 | 16.5369 | 17.0056 | 17.8537 | 18.6069 |
| 5 | 9.3516 | 11.1643 | 12.4014 | 13.3590 | 14.1624 | 14.8561 | 15.4736 | 16.0295 | 16.5389 | 17.0079 | 17.8566 | 18.6104 |
| 10 | 9.3520 | 11.1653 | 12.4030 | 13.3613 | 14.1654 | 14.8598 | 15.4781 | 16.0347 | 16.5450 | 17.0149 | 17.8655 | 18.6209 |
| 15 | 9.3526 | 11.1668 | 12.4057 | 13.3650 | 14.1703 | 14.8659 | 15.4855 | 16.0434 | 16.5551 | 17.0265 | 17.8799 | 18.6384 |
| 20 | 9.3536 | 11.1690 | 12.4095 | 13.3703 | 14.1772 | 14.8744 | 15.4959 | 16.0555 | 16.5693 | 17.0427 | 17.9001 | 18.6628 |
| 25 | 9.3547 | 11.1718 | 12.4143 | 13.3770 | 14.1860 | 14.8854 | 15.5093 | 16.0710 | 16.5874 | 17.0635 | 17.9261 | 18.6938 |
| 30 | 9.3562 | 11.1753 | 12.4202 | 13.3852 | 14.1968 | 14.8978 | 15.5255 | 16.0900 | 16.6095 | 17.0887 | 17.9575 | 18.7314 |

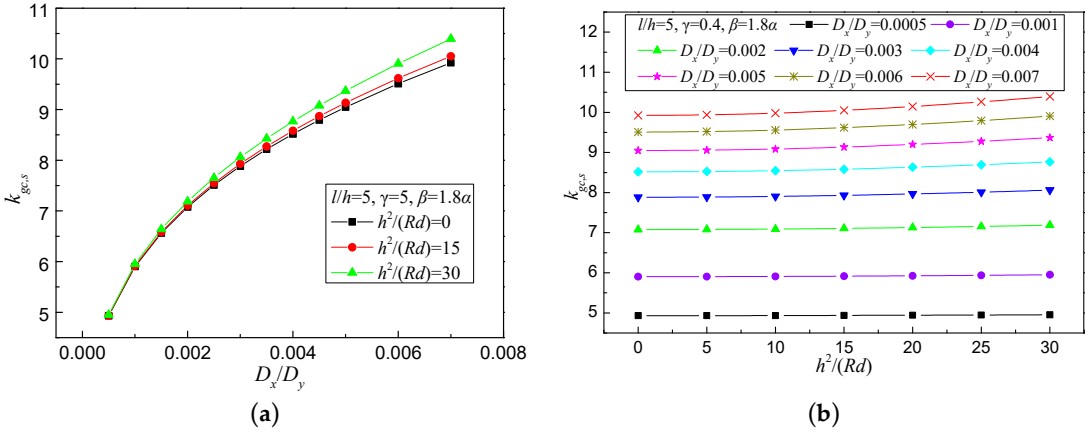

(a)　　　　　　　　　　　　　　　　　　　(b)

**Figure 4.** The effect of $D_x/D_y$ and $h^2/(Rd)$ on the global shear buckling coefficient $k_{g,s}$ for the four-edge simple support: (**a**) $D_x/D_y$ on $k_{g,s}$; (**b**) $h^2/(Rd)$ on $k_{g,s}$.

The first row of data for $h^2/(Rd) = 0$ in Tables 4–6 represents the global shear buckling coefficient of CSWs for straight girders. Through fitting of the first-row data in Tables 4–6, for CSWs with $0.0005 \leq \alpha \leq 0.007$, the global shear buckling coefficients $k_{g,s}$, $k_{g,f}$, $k_{g,fs}$ for straight girders can be estimated respectively by Equations (19) and (20).

For a four-edge simple support:

$$k_{g,s} = 36.8\alpha^{0.2648} \tag{19}$$

For a four-edge fixed support, or for two edges constrained by flanges fixed and the other two edges simply supported:

$$k_{g,f} = k_{g,fs} = 67.7\alpha^{0.2608} \tag{20}$$

For trapezoidal CSWs that are commonly used in actual bridges, the rigidity ratio $\alpha$ can be expressed as Equation (21).

$$\alpha = \frac{D_x}{D_y} = \frac{q^2 t^2}{2s(3a+c)d^2} \tag{21}$$

It is worth mentioning that the formulas for the global shear buckling stress $\tau_g^e$ (Equation (18)) and the global shear buckling coefficient $k_g$ (Equations (19) and (20)) proposed in this paper are slightly different from these proposed by previous researchers [8]. This is due to the fact that while previous researchers derived the formula for the global shear buckling stress $\tau_g^e$ the complete CSW was treated as an orthotropic plate with its length much larger than its width, and using a simplified deflection function, which is different from the function used in this paper.

## 2.3. Elastic Local Shear Buckling Stress of CSWs

Similar to straight girders, the local shear buckling of CSWs for curved girders can also be solved by analyzing a single flat panel constrained by adjacent panels and girder flanges under shear force. For straight and curved girder the same formulas apply for calculating the elastic local shear buckling stress of CSWs which are given by the classical plate buckling theory [3] as Equation (22):

$$\tau_l^e = k_l \frac{\pi^2 E}{12(1-v^2)} \left(\frac{t}{p}\right)^2 \tag{22}$$

where $k_l$ is the elastic local shear buckling coefficient of CSWs; $p$ is the maximum sub-panel width (maximum of flat panel width $a$ and inclined panel width $c$).

The elastic local shear buckling coefficient $k_l$ can be expressed as Equations (23)–(25).

For a four-edge simple support:

$$k_{l,s} = 5.34 + 4(p/h)^2 \tag{23}$$

For a four-edge fixed support:

$$k_{l,f} = 8.98 + 5.6(p/h)^2 \tag{24}$$

For the two edges constrained by flanges fixed and the other two edges simply supported:

$$k_{l,fs} = 5.34 + 2.31(p/h) - 3.44(p/h)^2 + 8.39(p/h)^3 \tag{25}$$

## 3. Finite Element Analysis

The central angle which depends on the radius of curvature $R$ and girder span $L$ is an important factor for HCGs. A linear FEA is carried out by ANSYS software (ANSYS 12.1, ANSYS Inc., Canonsburg, PA, USA, 2012) [32] to study the influence of $R$ and $L$ on the elastic shear buckling stress of CSWs for HCGs. Three geometries of CSWs that are commonly used in actual bridges are studied here (see Table 7).

**Table 7.** Studied geometries of CSWs.

| Model | $a$ (mm) | $b$ (mm) | $c$ (mm) | $d$ (mm) | $\theta$ (°) |
|---|---|---|---|---|---|
| CSW900 | 250 | 200 | 250 | 150 | 36.9 |
| CSW1200 | 330 | 270 | 336 | 200 | 36.5 |
| CSW1600 | 430 | 370 | 430 | 220 | 30.6 |

### 3.1. Finite Element Modeling

To reduce computational time, only one half of each girder is modeled here. The remainder (the right hand side part) is represented by adequate symmetry boundary conditions. The shell element (shell 181) is used to model the girders with CSWs. The finite element model is shown in Figure 5 and the boundary conditions are given in Table 8, where the edge AB is a roller support and the surface CDEF is a plane of symmetry. All girders are subjected to a concentrated load at the point G located in the plane of symmetry. Junction nodes of flanges and CSWs are constrained in the radial direction ($z'$-direction), the rotations around the $x'$ axis and $y$-axis to prevent lateral-torsion buckling.

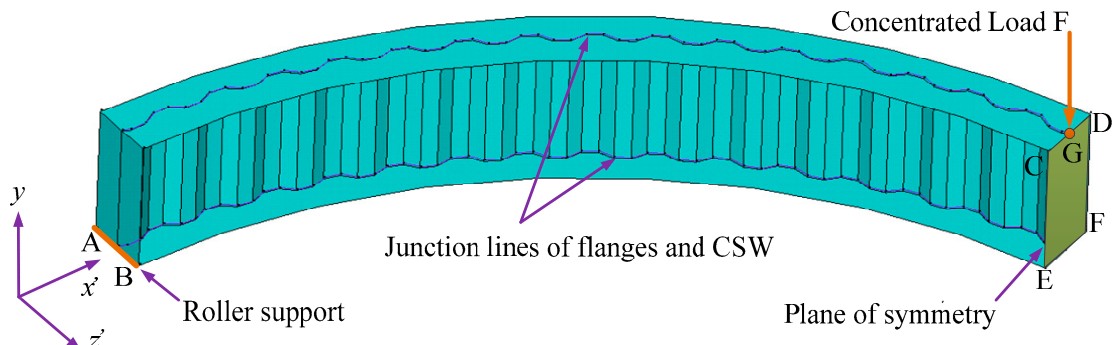

**Figure 5.** Load and boundary conditions of a curved girder with CSW.

**Table 8.** Boundary conditions of finite element models.

| Boundary | $\delta_{x'}$ | $\delta_y$ | $\delta_{z'}$ | $\theta_{x'}$ | $\theta_y$ | $\theta_{z'}$ |
|---|---|---|---|---|---|---|
| Edge AB | ○ | ● | ● | ● | ● | ○ |
| Surface CDEF | ● | ○ | ● | ● | ● | ● |
| Junction nodes | ○ | ○ | ● | ● | ● | ○ |

Note: ○: Free; ●: Restrained.

In this study, the radius of curvature $R$ is set at 30 m~150 m, the half span of the girders is set at 5$q$~25$q$, the width and the thickness of the flanges are 8$d$ and 100 mm respectively. The behavior of the stiffeners is assumed to be rigid. In addition, the number of elements per sub-panel is 6, as suggested by Eldib [2], and the element mesh size is $a/6$. The elastic modulus and poisson's ratio of steel are taken as 210,000 MPa and 0.3 respectively. Figure 6 represents three shear buckling modes of CSWs.

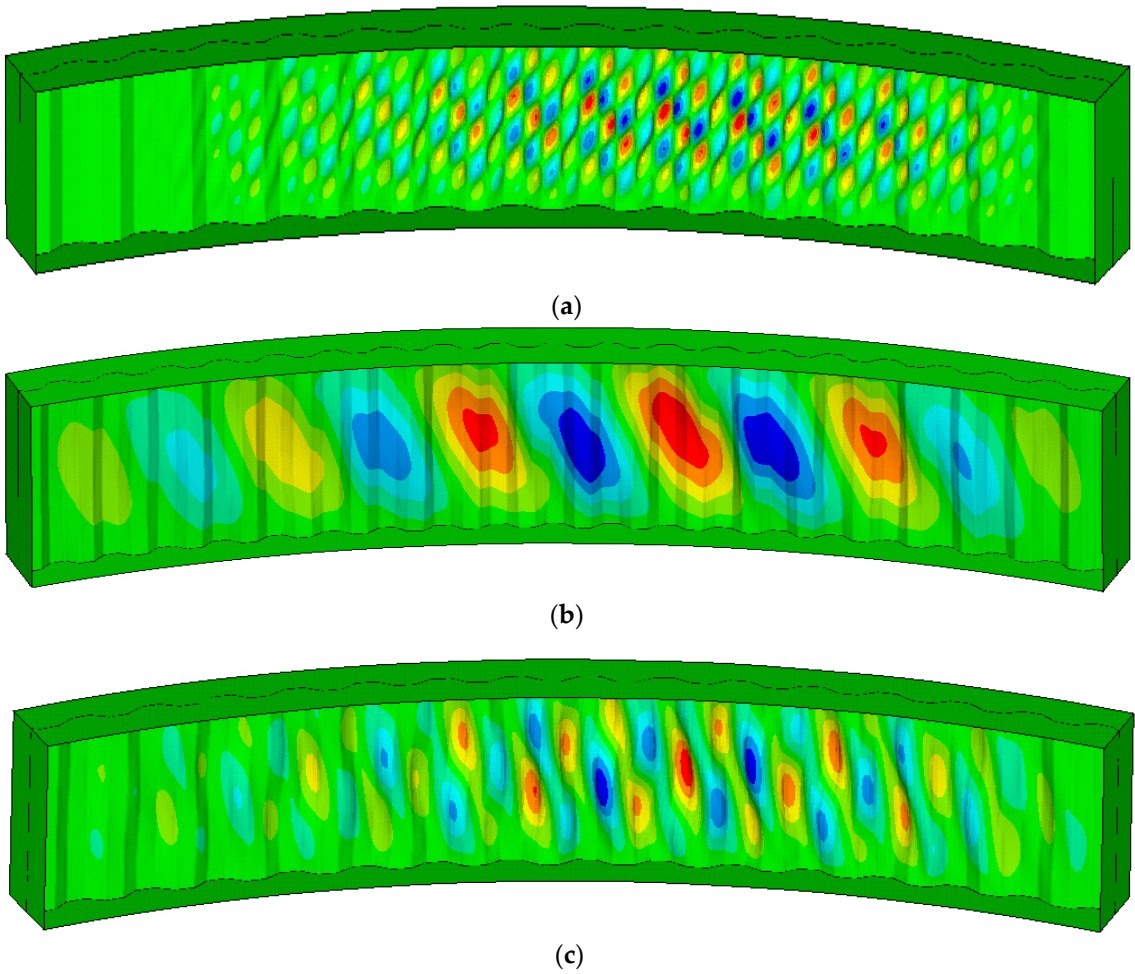

**Figure 6.** Three shear buckling modes: (**a**) Local shear buckling; (**b**) Global shear buckling; (**c**) Interactive shear buckling.

*3.2. Influence of the Radius of Curvature and Girder Span on the Elastic Shear Buckling Stress*

Theoretically, the radius of curvature $R$ has only a small influence on the elastic global shear buckling stress and has no influence on the elastic local shear buckling stress. Table 9 shows the elastic shear buckling stress of CSWs for different radii of curvature when $L/2 = 15q$. As we can see from Table 9, the difference of the FEA results between curved girders and straight girders with CSWs is very small which agrees with the theoretical expectations. Table 10 shows the elastic shear buckling stress of CSWs for different spans. It can be seen from Table 10 that the shear buckling stress decreases slightly with the increase of span and the buckling stress shows a converging trend with the increase of the span, which agrees with the theoretical expectations. In Tables 9 and 10, the theoretical stress $\tau_{cr}^{e}$ is the minimum of the elastic global and the local shear buckling stress. $\tau_{FEA}^{e}$ is the shear buckling stress obtained from FEA. Because the difference of elastic shear buckling stress of CSWs between curved girders and straight girders is very small, the shear buckling stress formulas for straight girders can be applied for curved girders.

**Table 9.** Elastic shear buckling stress of CSWs for different radii of curvature.

| Model | $h$ (mm) | $a/h$ | $t$ (mm) | $d/t$ | $R$ (m) | $\tau_{FEA}^{e}$ (Mpa) | $\tau_{cr}^{e}$ (Mpa) | $\tau_{FEA}^{e}/\tau_{cr}^{e}$ |
|---|---|---|---|---|---|---|---|---|
| CSW900 | 1250 | 0.2 | 8 | 18.8 | 30 | 1073.8 | 1069 | 1.00 |
| | | | | | 60 | 1073.6 | 1069 | 1.00 |
| | | | | | 90 | 1073.5 | 1069 | 1.00 |
| | | | | | 120 | 1073.5 | 1069 | 1.00 |
| | | | | | 150 | 1073.5 | 1069 | 1.00 |
| | | | | | 250 | 1073.5 | 1069 | 1.00 |
| | | | | | Straight | 1073.5 | 1069 | 1.00 |
| | 2500 | 0.1 | 12 | 12.5 | 30 | 1309.4 | 1064.4 | 1.23 |
| | | | | | 60 | 1308.5 | 1064.4 | 1.23 |
| | | | | | 90 | 1308.3 | 1064.4 | 1.23 |
| | | | | | 120 | 1308.3 | 1064.4 | 1.23 |
| | | | | | 150 | 1308.3 | 1064.4 | 1.23 |
| | | | | | 250 | 1308.2 | 1064.4 | 1.23 |
| | | | | | Straight | 1308.2 | 1064.4 | 1.23 |
| CSW1200 | 1650 | 0.2 | 8 | 25 | 30 | 655.7 | 613.5 | 1.07 |
| | | | | | 60 | 653.9 | 613.5 | 1.07 |
| | | | | | 90 | 654.3 | 613.5 | 1.07 |
| | | | | | 120 | 654.5 | 613.5 | 1.07 |
| | | | | | 150 | 654.6 | 613.5 | 1.07 |
| | | | | | 250 | 654.6 | 613.5 | 1.07 |
| | | | | | Straight | 644.7 | 613.5 | 1.05 |
| | 3300 | 0.1 | 12 | 16.7 | 30 | 933.6 | 916.2 | 1.02 |
| | | | | | 60 | 931.1 | 916.2 | 1.02 |
| | | | | | 90 | 931.1 | 916.2 | 1.02 |
| | | | | | 120 | 931.2 | 916.2 | 1.02 |
| | | | | | 150 | 931.2 | 916.2 | 1.02 |
| | | | | | 250 | 931.2 | 916.2 | 1.02 |
| | | | | | Straight | 928.8 | 916.2 | 1.01 |
| CSW1600 | 2150 | 0.2 | 10 | 22 | 30 | 596.2 | 564.6 | 1.06 |
| | | | | | 60 | 588.8 | 564.6 | 1.04 |
| | | | | | 90 | 595.4 | 564.6 | 1.05 |
| | | | | | 120 | 595.9 | 564.6 | 1.06 |
| | | | | | 150 | 596.1 | 564.6 | 1.06 |
| | | | | | 250 | 596.5 | 564.6 | 1.06 |
| | | | | | Straight | 596.8 | 564.6 | 1.06 |
| | 4300 | 0.1 | 14 | 15.7 | 30 | 712.2 | 675 | 1.06 |
| | | | | | 60 | 705.5 | 675 | 1.05 |
| | | | | | 90 | 708.6 | 675 | 1.05 |
| | | | | | 120 | 708.7 | 675 | 1.05 |
| | | | | | 150 | 708.7 | 675 | 1.05 |
| | | | | | 250 | 708.7 | 675 | 1.05 |
| | | | | | Straight | 708.7 | 675 | 1.05 |

**Table 10.** Elastic shear buckling stress of CSWs for different spans.

| Model | R (m) | h (mm) | a/h | t (mm) | d/t | L/2 | $\tau_{FEA}^e$ (Mpa) | $\tau_{cr}^e$ (Mpa) | $\tau_{FEA}^e/\tau_{cr}^e$ |
|---|---|---|---|---|---|---|---|---|---|
| CSW900 | 30 | 1250 | 0.2 | 8 | 18.8 | 5q | 1080.1 | 1069 | 1.01 |
| | | | | | | 10q | 1074.1 | 1069 | 1.00 |
| | | | | | | 15q | 1073.8 | 1069 | 1.00 |
| | | | | | | 20q | 1073.8 | 1069 | 1.00 |
| | | | | | | 25q | 1073.7 | 1069 | 1.00 |
| | 90 | 2500 | 0.1 | 12 | 12.5 | 5q | 1334.4 | 1064.4 | 1.26 |
| | | | | | | 10q | 1312.4 | 1064.4 | 1.23 |
| | | | | | | 15q | 1308.3 | 1064.4 | 1.23 |
| | | | | | | 20q | 1306.5 | 1064.4 | 1.23 |
| | | | | | | 25q | 1304.6 | 1064.4 | 1.23 |
| CSW1200 | 30 | 1650 | 0.2 | 8 | 25 | 5q | 656.3 | 613.5 | 1.07 |
| | | | | | | 10q | 655.9 | 613.5 | 1.07 |
| | | | | | | 15q | 655.7 | 613.5 | 1.07 |
| | | | | | | 20q | 655.6 | 613.5 | 1.07 |
| | | | | | | 25q | 655.6 | 613.5 | 1.07 |
| | 90 | 3300 | 0.1 | 12 | 16.7 | 5q | 947.1 | 916.2 | 1.03 |
| | | | | | | 10q | 933.2 | 916.2 | 1.02 |
| | | | | | | 15q | 931.1 | 916.2 | 1.02 |
| | | | | | | 20q | 930.1 | 916.2 | 1.02 |
| | | | | | | 25q | 929.0 | 916.2 | 1.01 |
| CSW1600 | 30 | 2150 | 0.2 | 10 | 22 | 5q | 597.3 | 564.6 | 1.06 |
| | | | | | | 10q | 596.5 | 564.6 | 1.06 |
| | | | | | | 15q | 596.2 | 564.6 | 1.06 |
| | | | | | | 20q | 596 | 564.6 | 1.06 |
| | | | | | | 25q | 596 | 564.6 | 1.06 |
| | 90 | 4300 | 0.1 | 14 | 15.7 | 5q | 712.2 | 675 | 1.06 |
| | | | | | | 10q | 710.3 | 675 | 1.05 |
| | | | | | | 15q | 708.6 | 675 | 1.05 |
| | | | | | | 20q | 707.5 | 675 | 1.05 |
| | | | | | | 25q | 705.7 | 675 | 1.05 |

## 4. Experimental Work

### 4.1. Geometric Dimensioning of Test Girders

Three single cell box girder test specimens are used for this study. The geometry of the curved box girders is shown in Figure 7. The length is 6.6 m and the radius of curvature is 8 m. The box girders have two or three intermediate steel diaphragms with a thickness of 8 mm. The results of material tests are given in Table 11.

**Table 11.** Results of the material tests for the steel used for the CSWs.

| Specimen | Number of Intermediate Diaphragms | Design Thickness (mm) | Actual Thickness (mm) | Yield Stress (MPa) | Ultimate Stress (MPa) |
|---|---|---|---|---|---|
| S1-t1d2 | 2 | 1 | 0.88 | 187.5 | 322.4 |
| S2-t2d2 | 2 | 2 | 1.74 | 263.9 | 364.9 |
| S3-t2d3 | 3 | 2 | 1.74 | 263.9 | 364.9 |

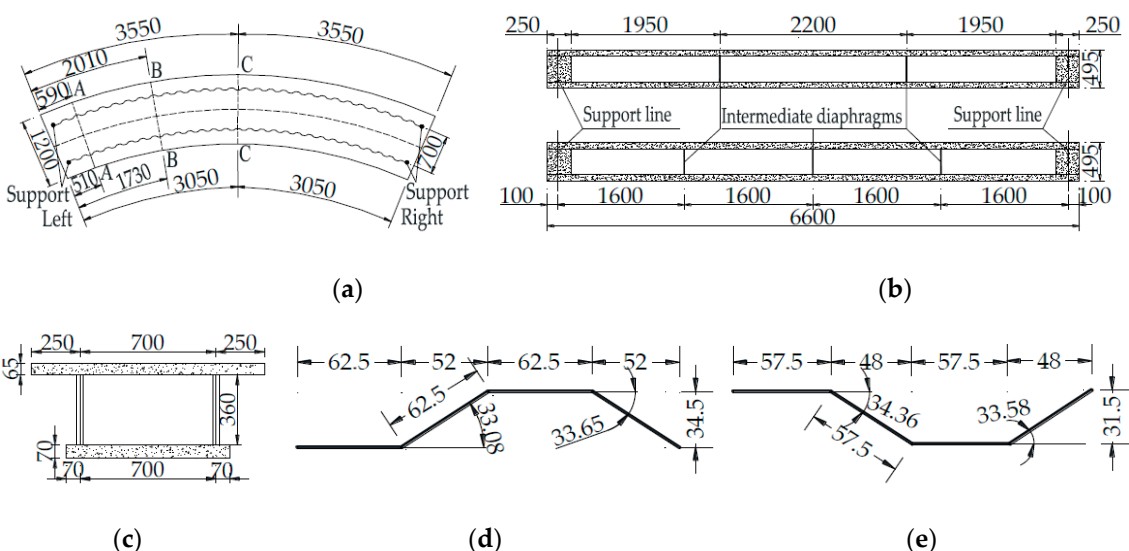

**Figure 7.** Geometry and dimensions of test girders with CSWs (unit: mm): (**a**) Plan view; (**b**) Elevation view; (**c**) Cross-section; (**d**) Outer CSW; (**e**) Inner CSW. Right: the right-hand side of test girders. Left: the left-hand side of test girders.

## 4.2. Loading Configurations and Measuring Devices

Figure 8 shows a schematic view of the loading configurations. Both midspan loading and three-point loading are applied in longitudinal direction, one and two-point loading are applied in radial direction. Figure 9 shows the arrangements of strain gauges on S1-t1d1 and S3-t2d3 and those on S2-t2d2 are similar. The size in Figures 8 and 9 express length of arc.

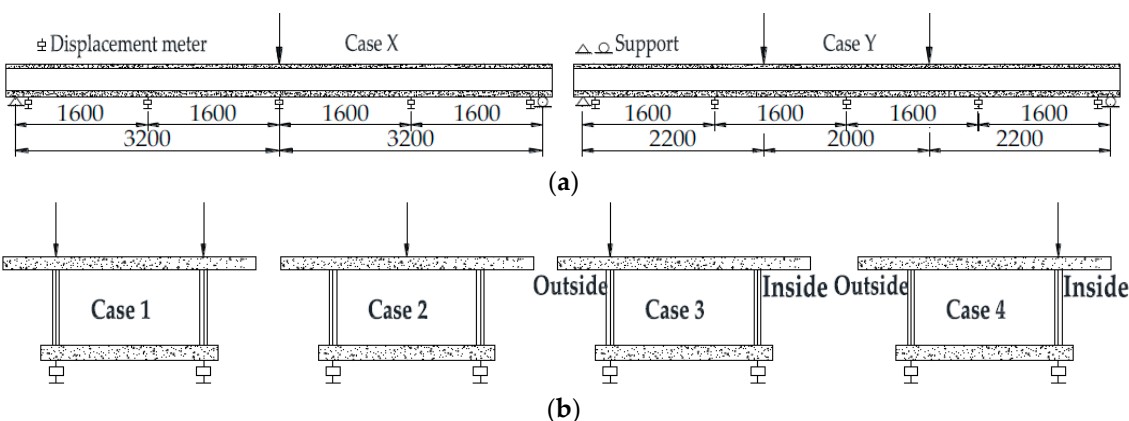

**Figure 8.** Schematic view of loading configurations: (**a**) Loading points applied in longitudinal direction; (**b**) Loading points applied in radial direction.

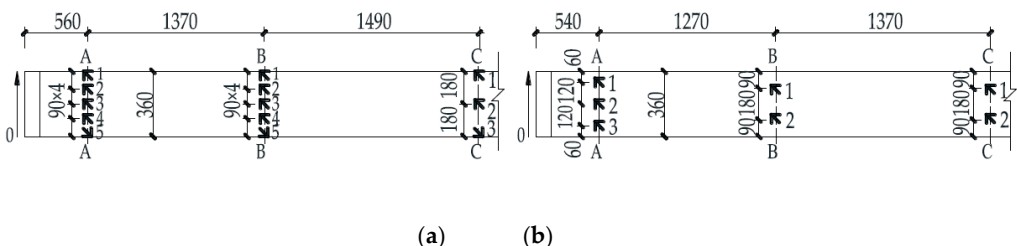

**Figure 9.** Arrangements of strain gauges on CSWs on S1-t1d2 and S3-t2d3: (**a**) Outer CSW; (**b**) Inner CSW.

Figure 10 shows a photograph of the test setup under the loading configuration of cases X and 2. Other loading configurations are similar. The details of the strain gauge configurations on the CSWs, the load transducers on the supports and the displacement transducers are shown.

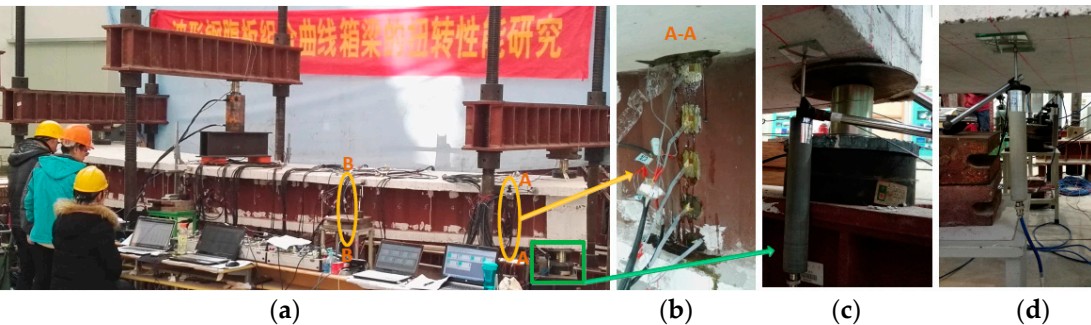

| (a) | (b) | (c) | (d) |

**Figure 10.** Photograph of the test setup: (**a**) Test setup; (**b**) Strain gauge configurations on CSWs; (**c**) Load transducers on supports; (**d**) Displacement transducers.

*4.3. Experimental Results*

4.3.1. Non-Destructive Test

First, a non-destructive test was carried out. In this test, a midspan loading was applied (see Figure 8 Case X).

As is shown in Figure 9, strain rosette gages were used to measure the strains in three directions. The shear strain $\gamma_w$ and the principle strain direction angle $\varphi$ can be calculated by Equations (26) and (27) respectively.

$$\gamma_w = \varepsilon_{0°} + \varepsilon_{90°} - 2\varepsilon_{45°} \tag{26}$$

$$\varphi = \frac{1}{2}\arctan\left(\frac{\gamma_w}{\varepsilon_{90°} - \varepsilon_{0°}}\right) \tag{27}$$

where $\varepsilon_{0°}$, $\varepsilon_{90°}$, $\varepsilon_{45°}$ are the strains in horizontal direction, vertical direction, and 45° direction respectively, as shown in Figure 11.

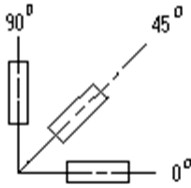

**Figure 11.** Schematic view of the strain rosette gage.

Figure 12 shows the measured shear strain distributions of the CSWs at sections A and B for specimen S3-t2d3. It can be found that the shear strain distributions are almost uniformly distributed along the direction of the web height. Because the effect of the concrete flanges, the shear strain near flanges shows some deviations. The principal strain direction angles of CSWs are given in Table 12. It can be seen that the direction angles are close to 45°, which indicates that CSWs for HCGs are almost in a pure shear stress state and barely carry axial forces. The results are similar to CSWs of straight girders [22].

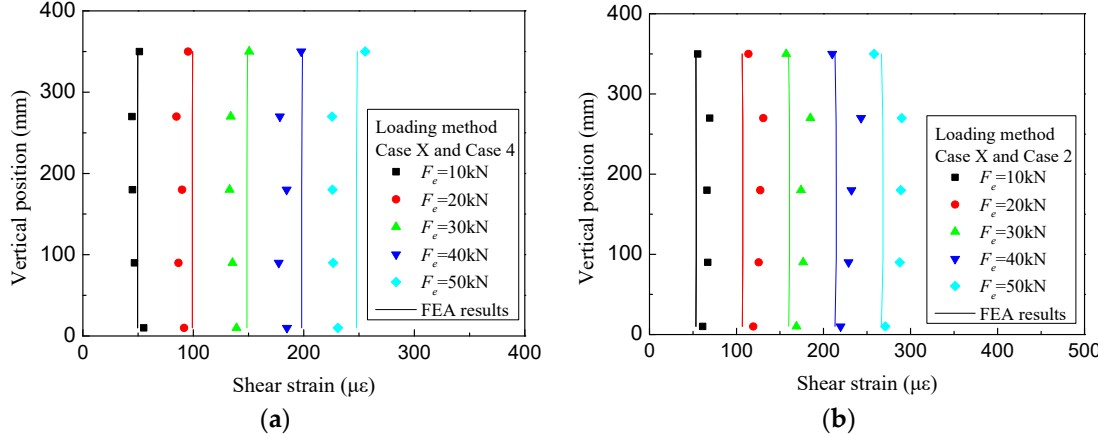

**Figure 12.** Shear strain distributions of CSWs for specimen S3-t2d3: (**a**) A section; (**b**) B section.

**Table 12.** Principle strain direction angles of CSWs.

| Specimens | Section | Loading Case | Vertical Position (mm) | $F_e$ (kN) | | | | |
|---|---|---|---|---|---|---|---|---|
| | | | | **10** | **20** | **30** | **40** | **50** |
| S2-t2d2 | B | Case 1 | 300 | 41.73° | 40.99° | 41.29° | 41.59° | 41.68° |
| | | | 180 | 43.17° | 44.04° | 44.40° | 44.59° | 44.60° |
| | | | 60 | 44.83° | 44.55° | 44.27° | 43.54° | 43.31° |
| S3-t2d3 | A | Case 4 | 350 | 40.13° | 40.77° | 40.57° | 40.24° | 40.49° |
| | | | 270 | 42.79° | 42.71° | 43.44° | 43.16° | 42.91° |
| | | | 180 | 44.24° | 44.54° | 44.77° | 44.77° | 44.71° |
| | | | 90 | 44.98° | 44.86° | 44.97° | 44.68° | 45.00° |
| | | | 10 | 44.46° | 43.80° | 43.93° | 43.60° | 43.71° |

According to structural mechanics, the total shear force of section A and B is 25 kN when applying 50 kN vertical load at midspan. The average shear strain in the same section is used to calculate $Q_w$, the shear force carried by the CSWs, which can be calculated by Equation (28) due to the uniformly distributed shear strain along the direction of the web height.

$$Q_w = G(\gamma_o + \gamma_i)th \tag{28}$$

where $\gamma_o$ and $\gamma_i$ are the average shear strain of the outer web and the inner web respectively. Table 13 shows the ratios of $Q_w$ to the total shear force $Q_{total}$ of specimen S3-t2d3, and the average ratio is 76%.

**Table 13.** Ratios of $Q_w$ to the total shear force $Q_{total}$ of specimen S3-t2d3.

| Specimens | Loading Case | Section | $Q_{total}$ (kN) | Test Results | | | | FEA Results | | | |
|---|---|---|---|---|---|---|---|---|---|---|---|
| | | | | $\gamma_o$ $(10^{-6})$ | $\gamma_i$ $(10^{-6})$ | $Q_w$ (kN) | $Q_w/Q_{total}$ | $\gamma_o$ $(10^{-6})$ | $\gamma_i$ $(10^{-6})$ | $Q_w$ (kN) | $Q_w/Q_{total}$ |
| S3-t2d3 | Case 2 | A | 25 | 327 | 44 | 18.8 | 75.1% | 344 | 34 | 19.2 | 76.6% |
| | Case 2 | B | | 289 | 98 | 19.6 | 78.3% | 267 | 111 | 19.1 | 76.6% |
| | Case 3 | A | 25 | 409 | −46 | 18.4 | 73.5% | 442 | −62 | 19.2 | 76.8% |
| | Case 3 | B | | 366 | 14 | 19.2 | 76.9% | 355 | 25 | 19.2 | 76.7% |
| | Case 4 | A | 25 | 233 | 128 | 18.3 | 73.1% | 248 | 130 | 19.1 | 76.5% |
| | Case 4 | B | | 188 | 194 | 19.3 | 77.3% | 181 | 197 | 19.1 | 76.4% |

### 4.3.2. Destructive Tests

(1) Shear Buckling Stress of CSWs

Based on the theoretical formulas (Section 2), the elastic global and local shear buckling stress of CSWs is shown in Table 14. It can be seen from Table 14 that the elastic global and local shear

buckling stresses of CSWs are larger than the shear yield stress of CSWs, that is to say, the occurrence of shear buckling occurs after the yielding of CSWs which meets the general safety criterion of bridge design [33]: yielding before local shear buckling, and local buckling before global shear buckling. In this way the full strength of the girder is mobilized. While local buckling cannot induce girder failure, global shear buckling can. After local buckling the load can still be increased.

**Table 14.** Shear Buckling Stress of CSWs.

| Specimen | CSW | Global Buckling Stress $\tau_{g,s}^e$ (MPa) | Local Buckling Stress $\tau_{l,s}^e$ (MPa) | Uniaxial Yield Stress (MPa) | Shear Yield Stress (MPa) |
|---|---|---|---|---|---|
| S1-t1d2 | Outer CSW | 1542 | 205.5 | 187.5 | 108.2 |
| | Inner CSW | 1347.9 | 241.9 | 187.5 | 108.2 |
| S2-t2d2/S3-t2d3 | Outer CSW | 2212.4 | 803.3 | 263.9 | 152.4 |
| | Inner CSW | 1934 | 945.9 | 263.9 | 152.4 |

(2)  Experimental Phenomena and Results

In the destructive tests, a three-point loading was applied (see Figure 8 Case Y and Case 3).

The final deformed shape of the test specimen S1-t1d2 at the end of testing is presented in Figure 13. The final buckling shapes are presented in Figures 14–16. From the experimental phenomena it is clear that local shear buckling occurred first in one of the corrugated panels. Then, with increasing load, the buckling propagated to adjacent panels. In the final state, the interactive shear buckling and the global shear buckling occurred in all specimens. It should be pointed out that the buckling of the CSWs occurred after the yielding of the CSWs which is in good agreement with the theoretical prediction.

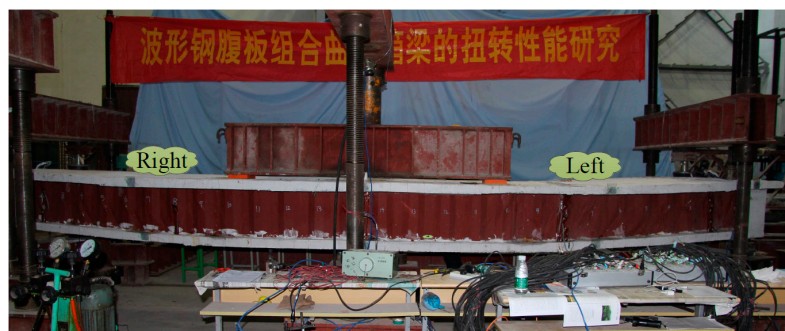

**Figure 13.** The final deformed shaped of specimen S1-t1d2.

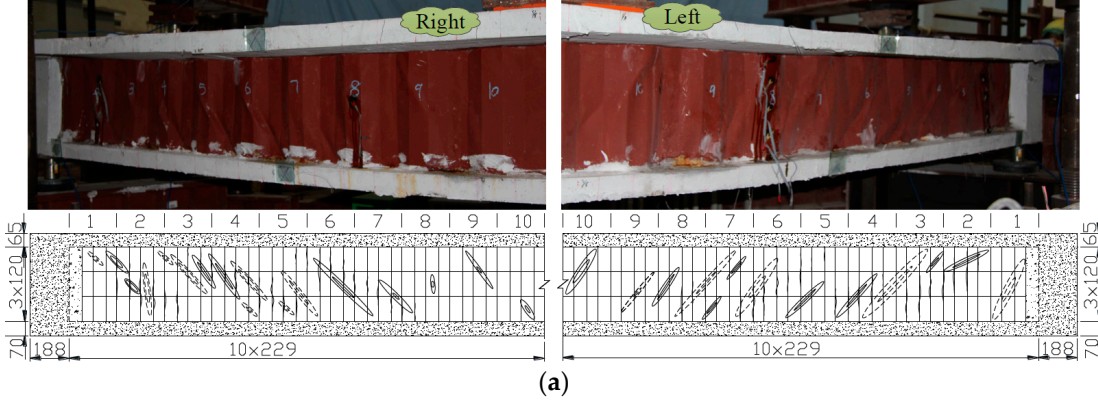

(**a**)

**Figure 14.** *Cont*.

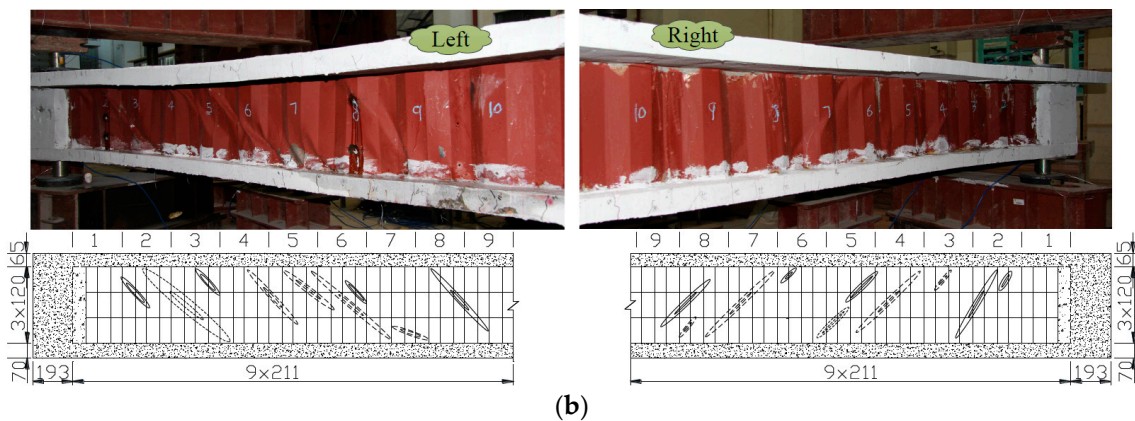

(**b**)

**Figure 14.** The final buckling shapes of CSWs for specimen S1-t1d2: (**a**) Outer CSW; (**b**) Inner CSW.

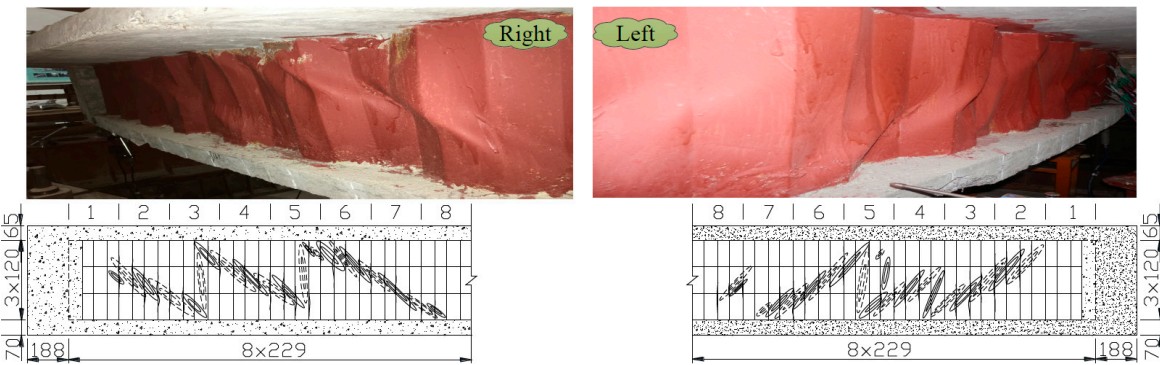

**Figure 15.** The final buckling shapes of outer CSW for specimen S2-t2d2. Right: the right-hand side of test girders. Left: the left-hand side of test girders.

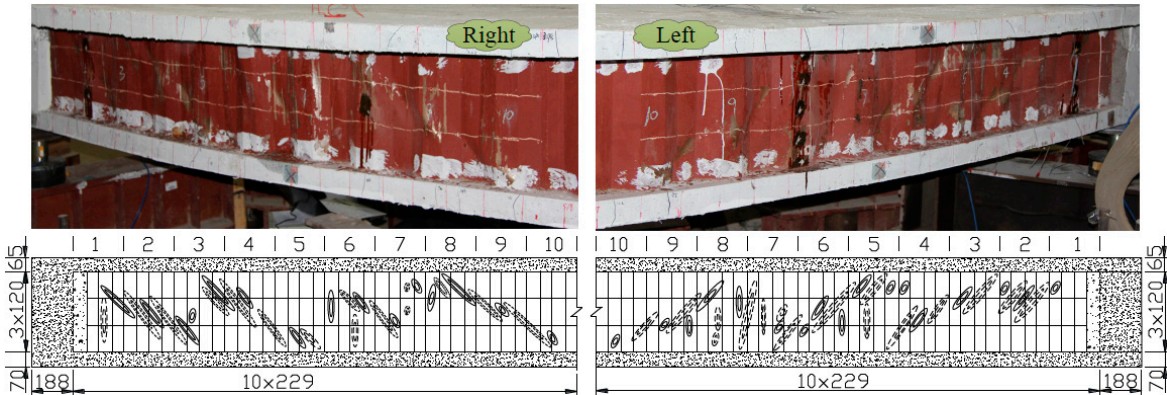

**Figure 16.** The final buckling shapes of outer CSW for specimen S3-t2d3. Right: the right-hand side of test girders. Left: the left-hand side of test girders.

Figure 17 shows the load-deflection curves of test girders. the curves do not give the displacement state at the stage of critical load due to failure of the displacement meters. The loads at which local buckling occurred are 65.9 kN, 140 kN and 147 kN for S1-t1d2, S2-t2d2, and S3-t2d3 respectively. The critical loads are 94.3 kN, 190 kN, 157.9 kN respectively. This shows that the box girders with CSWs have high post-buckling strength and can continue to resist the load after the occurrence of local shear buckling. Figure 17 also shows the load-deflection curves obtained from FEA results. It can be observed that the FEA results and the test results follow the same trend, and they are in good agreement at the early stage of loading. The agreement is reasonable to good at the later stage of the

loading considering the influence of simultaneous yielding and buckling of CSWs and stiffness loss due to concrete cracking.

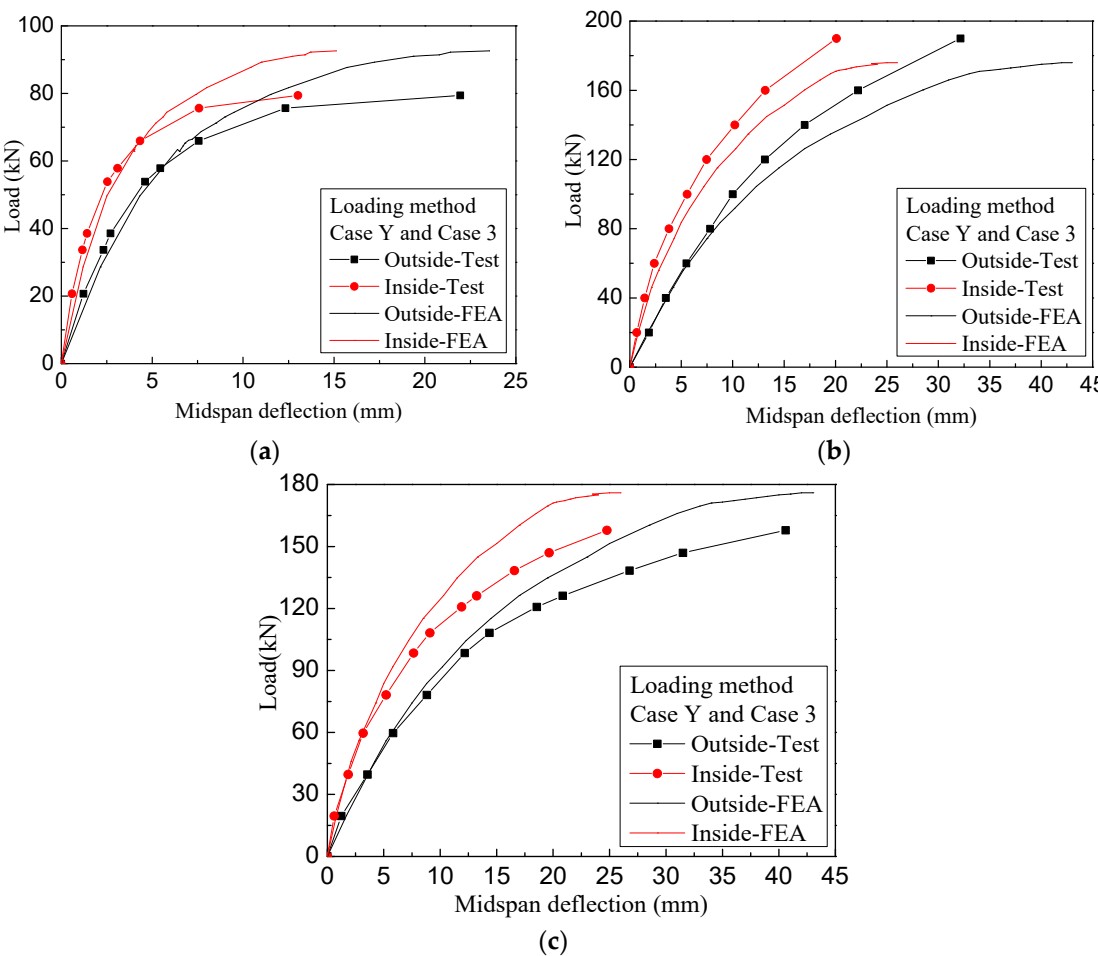

**Figure 17.** Load-deflection curves of test girders at mid-span section: (**a**) S1-t1d2; (**b**) S2-t2d2; (**c**) S3-t2d3.

## 5. Conclusions

In this paper, the shear capacity of CSWs for HCGs is theoretically, numerically and experimentally studied, and the following main conclusions can be drawn:

(1) The CSW in HCGs is treated as an orthotropic cylindrical shallow shell, and the analytical formula for the elastic global shear buckling stress is deduced by the Galerkin method. Simplified formulas for the global shear buckling coefficient $k_g$ for a four-edge simple support, for a four-edge fixed support, and for the two edges constrained by flanges fixed and the other two edges simply supported are proposed.

(2) A parametric study based on a linear buckling analysis is performed to analyze the effect of the curvature radius and girder span on the shear buckling stress. Analytical and numerical results show that the difference of shear buckling stress of CSWs between curved girders and straight girders is small, so the shear design formulas for straight girders can be applied for curved girders.

(3) Loading tests were performed on three curved box girders with CSWs. Similar to CSWs in straight girders, the shear strain distributions of CSWs in HCGs are almost uniform along the direction of the web height and the principal strain direction angles are close to 45°. For the three specimens, CSWs carry about 76% of the shear force. In the destructive test, shear buckling after yielding occurred in all specimens which is in good agreement with the theoretical prediction,

which means that the analytical formulas provide good predictions for the shear buckling stress of CSWs in HCGs and can be applied for design purposes.

**Author Contributions:** S.L. did the derivation of the global shear buckling stress of CSWs for HCGs and performed the finite element analysis and writing. H.D. and S.L. designed and performed the experiment. H.D., L.T. and W.D.C. did the review and editing.

**Funding:** This research was funded by the National Natural Science Foundation of China, grant number 51378106 and the China Scholarship Council. The financial support is gratefully acknowledged.

**Conflicts of Interest:** The authors declare no conflict of interest.

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
