# Peer review of "Shear Strength of Trapezoidal Corrugated Steel Webs for Horizontally Curved Girder Bridges"

_applsci, doi:10.3390/app9091942_

Round 1

Reviewer 1 Report

The reviewer would like to thank the author for the time and effort devoted to this research. It tries to achieve a deeper understanding of the behavior of corrugated steel webs in horizontally curved girder bridges.

The paper is original, since most of the research has been devoted to straight girders. 

The state-of-the-art section seems to be complete and updated. 

This reviewer would like to highlight that the paper it is very well organized, although it is deep and complex.  

The paper matches with the scope of the journal. 

It is important to remark that this reviewer has not been able to review all the mathematical formulae and expressions. Therefore, the authors MUST check again the text to remove any possible typo before publication.

This reviewer finds very interesting how the Section 2 has included data from real built bridges to simplify the practical formulation.

The conclusions can be drawn from the previous sections.

This reviewer can recommend the paper for publication, provided the following comments, that lead to minor corrections in the paper, are properly addressed:

1. Since the finite elements models are carried put using the ANSYS software, a reference to this code must be included in the reference section. 

2. In section 4.2. some photos should be added to clarify all the experimental work. 

3. As it was aforementioned, the authors MUST check all the mathematical expressions to remove any typo.

Author Response

Dear Reviewer,

We are thankful to you for pointing out some important modifications needed in the reports. We have thoughtfully taken into account these comments. The explanations of what we have changed in response to your comments are as follows:

Point 1: Since the finite element models are carried out using the ANSYS software, a reference to this code must be included in the reference section.

Response 1: Thank you for your suggestion. Considering your suggestion, we have added a reference relates to the ANSYS software in section 3, Page 11, Line 306.

Point 2: In section 4.2. some photos should be added to clarify all the experimental work.

Response 2: Thank you for your suggestion. Considering your suggestion, we have added a figure (Figure 10 including 4 experimental photographs) and a paragraph in section 4.2, Page 16.

Point 3: As it was aforementioned, the authors MUST check all the mathematical expressions to remove any typo.

Response 3: Thank you for your suggestion. We have carefully checked all the mathematical expressions to remove any typo.

We believe that the comments have been highly constructive and very useful to restructure the manuscript. We also believe that the new reference and photos included in the article really improved the quality of the manuscript.

We hope that all these changes fulfill the requirements to make the manuscript acceptable for publication in “Applied Sciences”.

Looking forward to hearing from you soon.

Thank you and best regards.

Yours sincerely,

Hanshan Ding; Sumei Liu; Luc Taerwe; Wouter De Corte; Xun Du

Reviewer 2 Report

The paper is very interesting, the comparison between experimental tests and theoretical model previsions are well conducted and well presented. The paper can be accepted in its present form.  

Author Response

Dear Reviewer,

Thank you very much for your comments.

Best regards.

Yours sincerely,

Hanshan Ding; Sumei Liu; Luc Taerwe; Wouter De Corte; Xun Du

Reviewer 3 Report

The manuscript has a very good structure and very well-presented.

Any comment is required, the manuscript can be accepted in the present form.

Author Response

(The authors gave the same response as above.)
